# Utilization and Regional Differences of In-Patient Services for Peripheral Arterial Disease and Acute Limb Ischemia in Germany: Secondary Analysis of Nationwide DRG Data

**DOI:** 10.3390/jcm11082116

**Published:** 2022-04-11

**Authors:** Matthias Trenner, Christoph Knappich, Bianca Bohmann, Simon Heuberger, Hans-Henning Eckstein, Andreas Kuehnl

**Affiliations:** 1Department for Vascular and Endovascular Surgery, Klinikum rechts der Isar, Technical University of Munich, 81675 Munich, Germany; mtrenner@joho.de (M.T.); ga45jum@mytum.de (B.B.); heuberger.simon@gmail.com (S.H.); hheckstein@web.de (H.-H.E.); a.kuehnl@tum.de (A.K.); 2Division of Vascular Medicine, St.-Josefs Hospital, 65189 Wiesbaden, Germany

**Keywords:** peripheral arterial disease, vascular surgical procedures, endovascular procedures, incidence, disease hotspot, burden of disease

## Abstract

Background: Peripheral arterial disease (PAD) and acute limb ischemia (ALI) pose an increasing strain on health care systems. The objective of this study was to describe the German health care landscape and to assess hospital utilization with respect to PAD and ALI. Methods: Secondary data analysis of diagnosis-related group statistics data (2009–2018) provided by the German Federal Statistical Office. Inclusion of cases encoded by the International Classification of Diseases (ICD-10) codes for PAD and arterial embolism or thrombosis. Construction of line diagrams and choropleth maps to assess temporal trends and regional distributions. Results: A total of 2,589,511 cases (median age 72 years, 63% male) were included, of which 2,110,925 underwent surgical or interventional therapy. Overall amputation rate was 17%, with the highest rates of minor (28%) and major amputations (15%) in patients with tissue loss. In-hospital mortality (overall 4.1%) increased in accordance to Fontaine stages and was the highest in patients suffering arterial embolism or thrombosis (10%). Between 2009 and 2018, the annual number of PAD cases with tissue loss (Fontaine stage IV) increased from 97,092 to 111,268, whereby associated hospital utilization decreased from 2.2 million to 2.0 million hospital days. Hospital incidence and hospital utilization showed a clustering with the highest numbers in eastern Germany, while major amputation rate and mortality were highest in northern parts of Germany. Conclusions: Increased use of endovascular techniques was observed, while hospital utilization to treat PAD with tissue loss has decreased. This is despite an increased hospital incidence. Addressing socioeconomic inequalities and a more homogeneous distribution of dedicated vascular units might be advantageous in reducing the burden of disease associated with PAD and ALI.

## 1. Introduction

Most likely due to an aging population and a growing burden of atherosclerotic risk factors, cardiovascular diseases are globally on the rise. Between 1990 and 2019, the number of affected people almost doubled to a total of 523 million with 18.6 million related deaths worldwide [1]. The upward movement is reflected in an increased number of people suffering from peripheral arterial disease (PAD), which has been rising globally from 164 to 202 million between 2000 and 2010, with the highest change rates observed in low- and middle-income countries [2]. While the largest share of people with PAD are asymptomatic or experience intermittent claudication, about 11% of all PAD patients suffer from chronic limb-threatening ischemia (CLTI), corresponding to a prevalence of 1.3% in a population aged ≥40 years [3]. The annual amputation rate rises with increasing PAD categories and was found to be 60% for Rutherford class 6 patients [4].

Aside from the individual patient’s suffering, PAD poses an enormous strain on health care systems, particularly due to the abundance of associated comorbidities. Over the past years, different approaches have been applied to further characterize the cohort of PAD patients in Germany. The multicentric prospective GermanVasc cohort study found that >1/3 of PAD patients suffered severe systemic disease, with 3% exhibiting a constant threat to life [5]. Health care-related expenditures for patients with PAD are significantly higher compared to those without [6]. A retrospective health insurance claims data analysis of patients insured by one of the largest insurance providers in Germany found that disease related reimbursement costs rose by 31% between 2008 and 2016 [7]. Mean costs were shown to rise with higher Rutherford classes [4]. Annual costs were particularly high in PAD patients after they underwent major amputation [8]. While PAD generally describes a chronic condition, acute-on-chronic arterial occlusion may occur and (besides other causes including embolism, peripheral arterial aneurysm, dissection, injury, vasculitis) is causative in a major part of cases presenting as acute limb ischemia (ALI) [9,10,11]. Therefore, ALI, evolving on the basis of PAD and often requiring urgent or emergent treatment, further challenges health care systems.

The present study was conducted to delineate the utilization of the German health care system to treat PAD and ALI patients with respect to different treatment modalities, temporal trends, and regional distribution. Analysis of time course and regional distribution of PAD burden will support health policy makers to improve planning and control the provision of regional health care services.

## 2. Materials and Methods

The principal methods of this study have been applied previously by our research group and are described elsewhere [12,13,14,15,16].

### 2.1. Data Extraction

Diagnosis-related group (DRG) statistics data of the German Federal Statistical Office (GFSO) from 2009–2018 were extracted for secondary data analysis.

As virtually all (excluding military and psychiatry services) case related data of patient hospital episodes are obliged to be submitted to the GFSO by the German Institute for the Hospital Remuneration System (InEK) by §21 Hospital Finance Act (Krankenhausentgeltgesetz), this analysis by approximation was a full survey of the German population and hospitals. Due to the completeness of the data, imputation of missing data was not necessary. Follow-up of each DRG case covered the period from hospital admission to discharge. Legal basis of the use of DRG statistics for secondary data analysis is the German Federal Statistics Act (§3a; §16). The study protocol was approved by the ethics committee of the Medical Faculty, Technical University of Munich (Reference 21/16 S). The study was conducted in compliance with good practice of secondary data analysis [17] and the manuscript was written in accordance to the STROSA guideline (2nd version) [18].

Controlled remote data processing was used to access data on the servers of the GFSO according to data protection rules [12,13,14,15].

Data of the GFSO and Federal Institute for Research on Building, Urban Affairs, and Spatial Development (BBSR; INKAR database) were linked. Sex and age-specific population, district type (settlement structure, defined by BBSR), and length of hospital stay were linked. To identify the treating hospital, the institutional identification (“Institutskennung”) was linked to individual hospital addresses (conducted by the GFSO). This allowed for the calculation of distances between the patients’ residencies (center of postal code area) and actual hospital location. Data protection regulations were realized by GFSO employees. Type of treating department was directly adopted from the GFSO DRG database.

### 2.2. Study Population

Administrative codes used in this analysis are listed in Appendix A. All DRG cases encoded by International Classification of Diseases (ICD-10) codes for PAD or arterial embolism or thrombosis (AET) were included (Figure 1). Disease stages were operationalized using the ICD-10 code (Appendix A). In case of disparities between primary and secondary diagnoses, the main diagnosis was used for classification. As it was not possible to distinguish between arterial embolism and thrombosis, a relevant proportion of ALI cases will be attributable to causes other than PAD (e.g., cardiac or arterio-arterial embolism, peripheral arterial aneurysm, vasculitis, injury). To assess patients who underwent specific treatment, all cases encoded by German operation procedure codes (OPS) for open-surgical procedures (e.g., endarterectomy, various bypasses, surgical revisions), endovascular procedures (e.g., percutaneous transluminal angioplasty (PTA), stenting, implantation of covered stents), procedures to treat ALI (e.g., embolectomy, thrombectomy, thrombolysis), or amputations (e.g., forefoot amputation, below or above the knee amputation) were included. Cases transferred to another hospital without treatment were excluded.

All analyses refer to cases rather than patients as a link to individual data was not feasible.

### 2.3. Patient Characteristics, Hospital Incidence and Utilization, Outcomes

To report time trends in hospital incidence, hospital utilization, and treatment modalities, data from 2009–2018 were used. As the study aimed at describing the absolute utilization of vascular services for medical demand planning or hospital planning, no age and sex standardization was performed regarding incidence, while it was performed for the analysis of mortality.

To demonstrate regional distribution of hospital incidence, hospital utilization, and outcomes, choropleth maps were constructed. Thereby, aggregated data from 2018 for each administrative district was illustrated in a defined color or color intensity. Moran’s I statistic was performed to assess for spatial autocorrelation. Hotspot statistics were performed using Getis-Ord Gi * statistics, which is a method for analyzing the local values of a dataset. The resulting Z-values and *p*-values allow for determination where regions with high or low values tend to form spatial clusters [19,20]. With this analysis, each spatial value is evaluated considering the neighboring values. Regions with a positive *Z*-value and *p*-values ≤ 0.1 were defined as hot spots (99% hot spot: *p* < 0.01; 95%: *p* = 0.01–0.05; 90% *p* = 0.051–0.1) while regions with a negative *Z*-value and with respective *p*-values were classified as cold spots.

### 2.4. Statistical Analyses

Absolute numbers and percentages are given for categorical variables (i.e., sex, comorbidities, treating department, minor and major amputation, in-hospital death, endovascular, and open surgical procedures). Medians with first (Q1) and third quartile (Q3) are given for continuous variables (i.e., age, modified Elixhauser Score, distance from residency to hospital, length of hospital stay, annual sum of days in hospital).

As the database is a full survey, calculation of confidence intervals or *p* values is not necessary.

Controlled remote data processing was carried out using SAS (version 9.4M4 for Microsoft Windows, Copyright 2016, SAS Institute Inc., Cary, NC, USA). Analyses and graphics were performed using R (version 4.0.5, the R Foundation, www.r-project.org (accessed on 11 March 2022)) and Microsoft Excel (Version 2013, Microsoft Corporation, Redmond, WA, USA).

## 3. Results

### 3.1. Baseline Characteristics

Between 2009 and 2018, more than 2.5 million people were treated for peripheral arterial disease in Germany, whereby 2.1 million underwent interventional or surgical therapy (Table 1). With 39%, the highest proportion of cases was treated by general surgical departments, followed by vascular surgery (29%) and internal medicine departments (15%). The average length of stay was nine days if patients received interventional or surgical treatment and six days if treated conservatively. Overall mortality was 4.1%.

### 3.2. Characteristics of Interventionally or Surgically Treated Cases

Most patients undergoing interventional or surgical treatment ranged in Fontaine stage IIb, followed by stages IV and III (Table 2). The average patient age rose with Fontaine stages from 68 years (Fontaine stages I/IIa) to 76 years (Fontaine stage IV). The proportion of patients suffering chronic ischemic heart disease (30%), diabetes (45%), and renal disease (39%) was highest in Fontaine stage IV patients.

While vascular surgery and general surgery units predominantly treated cases ranging in Fontaine stages IIb upward, the highest proportion of cases treated in internal medicine units ranged from stages I to IIa.

Median length of stay increased from two days in Fontaine stages I or IIb to 14 days in stage IV. Similarly, in-hospital mortality increased with Fontaine stages from 0.1 to 5.8%. The highest mortality (10.1%) was found for patients suffering arterial embolism.

### 3.3. Procedural Characteristics of Interventionally or Surgically Treated Cases

Mild PAD (Fontaine stages I–IIb) was predominantly treated by above knee balloon angioplasty (64%) with below knee balloon angioplasty only being performed in 10% (Table 3).

CLTI (Fontaine stages III and IV) involved a higher proportion of below knee balloon angioplasty (26%).

Regarding open surgical procedures, mild PAD was most commonly treated with femoral endarterectomy (20%). Bypass surgery was performed in 12% with the distal anastomosis below the knee in 2.9%.

For treatment of CLTI, bypass surgery was applied in 23%. In 14% of cases, the distal bypass anastomosis was situated below the knee.

In the presence of tissue loss (Fontaine stage IV), an amputation was necessary in 46%, of which two thirds (67%) were minor amputations.

Arterial embolism or thrombosis was most frequently treated by open surgical above knee procedures (28%), followed by balloon angioplasty above (21%) and below the knee (11%).

### 3.4. Temporal Trends of Cases, Hospital Utilization, and Procedures

Between 2009 and 2018, the overall number of PAD and ALI cases increased from 232,302 to 259,572 (Figure 2A). Increases were observed for cases in all but Fontaine stage III, with the largest gain (15%) in cases with tissue loss.

Regarding arterial embolism or thrombosis, there was a rise from 33,619 to 40,624 cases.

Hospital utilization was measured in hospital days and decreased for PAD across all Fontaine stages (Figure 2B).

Regarding ALI, a rise of arterial embolism or thrombosis was found from 536,350 to 617,025 days.

With respect to applied procedures, the largest increase was found for endovascular procedures, while a decline in bypass surgery was observed (Figure 2C).

### 3.5. Regional Distribution of Cases, Hospital Utilization, and Outcomes

Choropleth maps were constructed to assess regional distributions for the year 2018. Clustering in northeastern German districts was observed for cases, hospital days, and procedures per 100,000 inhabitants (Figure 3).

The Getis-Ord Gi * analysis revealed 30 administrative regions as hot spots of PAD and ALI incidence, of which 11 were classified as 99% hot spots (*p*-level < 0.01: Coburg, Hof, Kronach, Nordsachsen, Anhalt-Bitterfeld, Saalekreis, Schmalkalden-Meiningen, Hildburghausen, Ilm-Kreis, Sonneberg, and Saalfeld-Rudolstadt), 13 as 95% hot spots (*p*-level = 0.01–0.05: Fulda, Kulmbach, Elbe-Elster, Potsdam-Mittelmark, Mecklenburgische Seenplatte, Leipzig country, Magdeburg, Harz, Wittenberg, Suhl, Wartburgkreis, Gotha, and Saale-Orla-Kreis), and six as 90% hot spots (*p*-level = 0.051–0.1: Dahme-Spreewald, Spree-Neiße, Vorpommern-Greifswald, Leipzig city, Dessau-Roßlau, and Stendal). Groß-Gerau (*p* = 0.042) and Böblingen (*p* = 0.045) were identified as 95% cold spots of incidence.

Three administrative regions appeared to be 99% hot spots in terms of hospital days per 100,000 inhabitants (*p*-level < 0.01: Hof, Kronach, and Hildburghausen), 12 as 95% hot spots (*p*-level = 0.01–0.05: Coburg, Kulmbach, Dahme-Spreewald, Anhalt-Bitterfeld, Saalekreis, Kyffhäuserkreis, Schmalkalden-Meiningen, Gotha, Ilm-Kreis, Sonneberg, Saalfeld-Rudolstadt, and Saale-Orla-Kreis), and 10 as 90% hot spots (*p*-level = 0.051–0.1: Essen, Fulda, Saarlouis, Elbe-Nelster, Spree-Neiße, Erzgebirgskreis, Bautzem, Meißen, Nordsachsen, and Suhl). Ten regions were identified as cold spots (95%; *p*-level = 0.01–0.05: Böblingen, and Breisgau-Hochschwarzwald; 90%; *p*-level = 0.051–0.1: Darmstadt-Dieburg, Offenbach, Calw, Enzkreis, Emmendingen, Schwarzwald-Baar-Kreis, Reutlingen, and Biberach).

Eleven regions were classified as hot spots in terms of the number of procedures per 100,000 inhabitants (99%; *p*-level < 0.01: Coburg, Hof, Kronach, Hildburghausen, Sonneberg, and Saalfeld-Rudolstadt; 95%; *p*-level = 0.01–0.05: Kulmbach, and Saale-Orla-Kreis; 90%; *p*-level = 0.051–0.1: Lichtenfels, Nordsachsen, and Saalekreis), while three were classified as 90% cold spots (*p*-level = 0.051–0.1: Fürth, Augsburg, and Donau-Ries).

Regarding major amputations and mortalities, a rise was observed toward northern regions of Germany (Figure 4).

With respect to the specialty of the treating center, it was found that PAD patients were predominantly treated by surgical divisions across Germany (Appendix A). Particularly for PAD in Fontaine stages I and II, regions exist where PAD patients are treated predominantly by other specialties (Appendix A). For the treatment of CLTI (Appendix A) or ALI (Appendix A), only in a few areas were patients treated predominantly by non-surgical specialties.

Although it has to be taken into account that even large (but not independent) vascular units might be administratively coded as general surgical units, it was found that regions where PAD patients are treated predominantly by dedicated vascular (i.e., vascular surgery or angiology) units are situated, especially in western Germany (Appendix A). In most of Germany, these patients are treated predominantly by less specialized (e.g., cardiology, internal medicine, general surgery) departments.

## 4. Discussion

The present study represents a full survey to assess the German health care landscape with respect to patient characteristics, time trends, and regional distributions to treat PAD and ALI.

It was demonstrated that patients with PAD often suffer multiple comorbidities and are at high risk for amputation (17%) and in-hospital death (4.1%). The complexity of patients increases with the severity of PAD, as higher Fontaine stages are associated with more comorbidities, higher amputation rates, and higher mortality. Between 2009 and 2018, an increase in PAD cases was observed, applying especially to the most demanding group of patients with PAD involving tissue loss (Figure 2A). Different studies have found a rise in PAD prevalence over the past years [2,7,21]. Apart from an elevated life expectancy, this might be due to an increasing prevalence of diabetes mellitus [22].

Notably, between 2014 and 2015, a notch was found in the curve of cases for PAD in Fontaine stage III with a contrary movement for Fontaine I/IIa cases (Figure 2A). Coding errors might be the reason as the coding system was changed in 2015 (Appendix A) and some cases in Fontaine stage III might have been coded as Fontaine stage I/IIa.

Despite a remarkable increase in PAD cases with tissue loss, the associated hospital utilization declined (Figure 2B). On one hand, this might be explained by a progressive use of endovascular techniques (Figure 2C), typically requiring shorter hospital stays. On the other hand, hospitals are increasingly confronted with economic pressure and are therefore inclined to limit hospital stays to a minimum.

Notably, the upward trend of PAD cases with tissue loss was not paralleled by an increase in amputations (Figure 2C). A retrospective study of German health insurance claims data more specifically demonstrated that major amputations decreased by 15%, while minor amputations rose by 28% between 2008 and 2016 [7]. This observation was confirmed by an observational study on administrative data provided by the Statistical Office of the European Union (EUROSTAT) [23]. The stable number of overall amputations might be explained by improvements in surgical and especially in endovascular techniques, allowing for revascularization of very distal below knee pathologies.

In this study, regional clustering was found for hospital incidence, hospital utilization, and procedures toward northeastern Germany (Figure 3). Previous studies have reported on the clustering of other conditions related to atherosclerotic disease such as carotid artery disease or abdominal aortic aneurysm disease [24,25,26]. A pooled data analysis from the German Health Update showed variations in lifetime prevalence of major cardiovascular disease disfavoring federal states in eastern Germany [27]. Clustering is probably influenced by the distribution of risk factors, environmental factors, socioeconomic factors, and other determinants of variation (e.g., demand, supply, data inaccuracy) [24]. In the case of PAD, a higher proportion of ≥65 year old individuals [26], higher prevalence of tobacco smoking [28], and lower individual income [29] in northeastern German federal states might be important drivers. Therefore, addressing socioeconomic inequalities is likely to reduce the prevalence of PAD and the associated burden of disease. Hospitals and authorities of regions identified as hot spots in the Getis-Ord Gi * statistics should use the results as inducement for further investigation regarding hospital planning and the implementation of vascular units.

Remarkably, there seem to be large parts of the country—especially in the eastern parts of Germany, where the incidence for PAD is highest—in which PAD and ALI patients are predominantly treated by other than dedicated vascular units (Appendix A). This observation has to be interpreted with caution, as even large vascular units might be integrated into general surgical units and coded accordingly. Nevertheless, to encounter the increasing burden of disease associated with PAD and ALI, it might be advantageous to augment the number of vascular units, distribute them more homogenously across the country, and direct patient flows toward them.

### Limitations

This study had a number of limitations, which in part have been previously reported [12,16].

The underlying dataset contained administrative data with hospital remuneration as the primary purpose. Despite regular controls by the Medical Review Board of the Statutory Health Insurance Funds, underreporting of comorbidities not affecting hospital remuneration with falsely low comorbidity rates cannot be ruled out. Nevertheless, hospital incidence and death were considered reliable outcomes. As there was no individual patient identifier, double counts could not be excluded, however, the main purpose of this study was to evaluate hospital utilization. For this question, double counts of individual patients do not seem relevant. The same accounts for follow-up after hospital discharge.

Coding errors or intentional up-coding cannot be excluded. However, due to the high number of cases, their impact on the whole cohort was considered negligible. Furthermore, adequacy of coding is regularly checked by the Medical Review Board of the Statutory Health Insurance Funds. Although a relevant proportion of ALI cases may not be attributed to an underlying PAD, this study aimed to describe the overall utilization of peripheral vascular treatment services in Germany. As the study design did not allow for disentanglement of arterial thrombosis from embolism, a relevant proportion of included ALI cases will not be associated with PAD. Type of treating department was directly adopted from the DRG data. Due to differences in local organization structures, even large (but not independent) vascular units might be administratively coded as general surgical units (e.g., department for general, visceral, and vascular surgery). Furthermore, it was not possible to extract information on symptom status for ALI cases (e.g., Rutherford classification). As only one ICD-10 code existed for PAD staged Fontaine I and IIa between 2005 and 2015 (Appendix A), it was not possible to categorize symptoms by intermittent claudication and CLTI, respectively.

Due to data protection regulations, Moran’s I statistics and Getis-Ord Gi * statistics could not be performed for all clinical outcomes.

Despite the listed limitations, the present study represents a full survey comprising virtually all patients suffering PAD and AET and utilizing in-patient services in Germany between 2009 and 2018, therefore minimizing the risk of selection bias.

Differences regarding incidence and mortality of PAD and AET are most likely due to the different age structure of the regional population. As the main purpose of this paper was the absolute (not the relative) burden of disease and hospital utilization, age and sex standardization would not have been appropriate.

## 5. Conclusions

The present study underlines the complexity of PAD and ALI patients as they are afflicted with various comorbidities and high mortality and morbidity. Therefore, PAD poses a major challenge to the German health care system. It was shown that the hospital incidence of PAD with tissue loss has been increasing. Nevertheless, and possibly due to an increased use of endovascular techniques, hospital utilization to treat PAD with tissue loss has decreased. Clustering of PAD and ALI was found disfavoring northeastern Germany. Addressing socioeconomic inequalities and a more homogeneous distribution of dedicated vascular units might be advantageous in reducing the burden of disease associated with PAD and ALI.

## Figures and Tables

**Figure 1 jcm-11-02116-f001:**
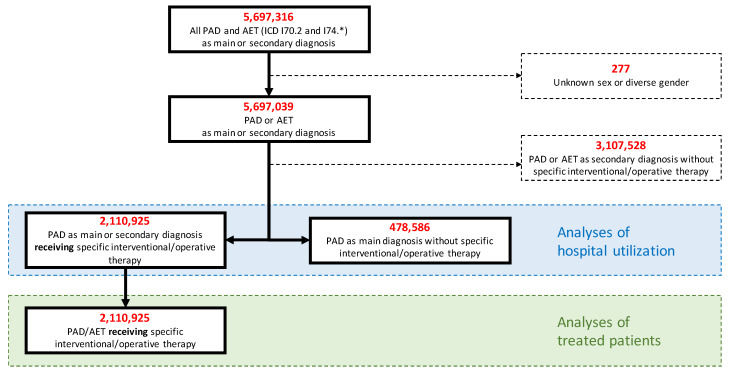
Flowchart. PAD indicates peripheral arterial disease; AET, arterial embolism or thrombosis.

**Figure 2 jcm-11-02116-f002:**
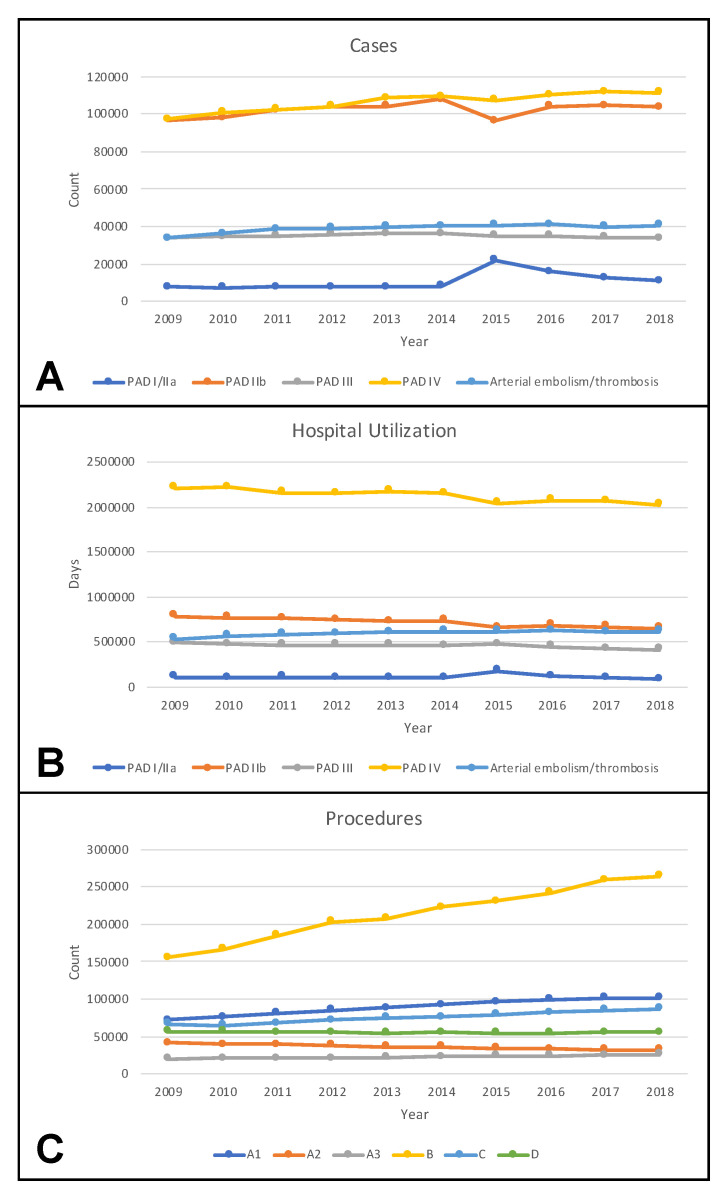
Temporal trends of cases (**A**), hospital utilization (**B**), and procedures (**C**) in Germany between 2009 and 2018. PAD indicates peripheral arterial disease with respective Fontaine stage; A1, local reconstructions including endarterectomy and patch angioplasty; A2, bypass procedures; A3, surgical revisions; B, balloon angioplasty and stenting; C, surgical and endovascular procedures to treat acute ischemia including (rotational) thrombectomy and thrombolysis; D, amputations.

**Figure 3 jcm-11-02116-f003:**
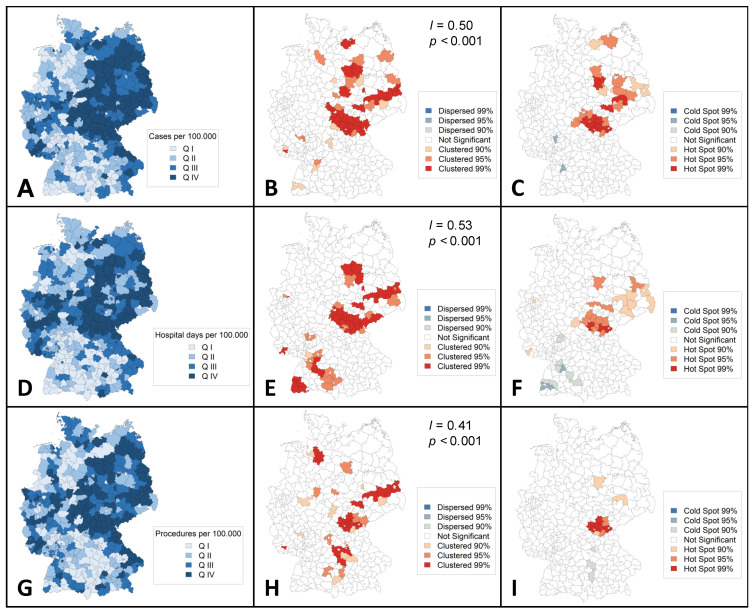
Regional distribution of cases and hospital utilization (aggregated 2018) in Germany. Choropleth maps showing regional distribution of cases (**A**–**C**), hospital days (**D**–**F**), and procedures (**G**–**I**), all per 100,000 inhabitants. Moran’s I statistics (**B**,**E**,**F**) were performed to assess for spatial autocorrelation. Getis-Ord Gi * statistics (**C**,**F**,**I**) were performed to identify hotspots. Q indicates quartile; *I*, Global Moran’s I; *p*, *p*-value.

**Figure 4 jcm-11-02116-f004:**
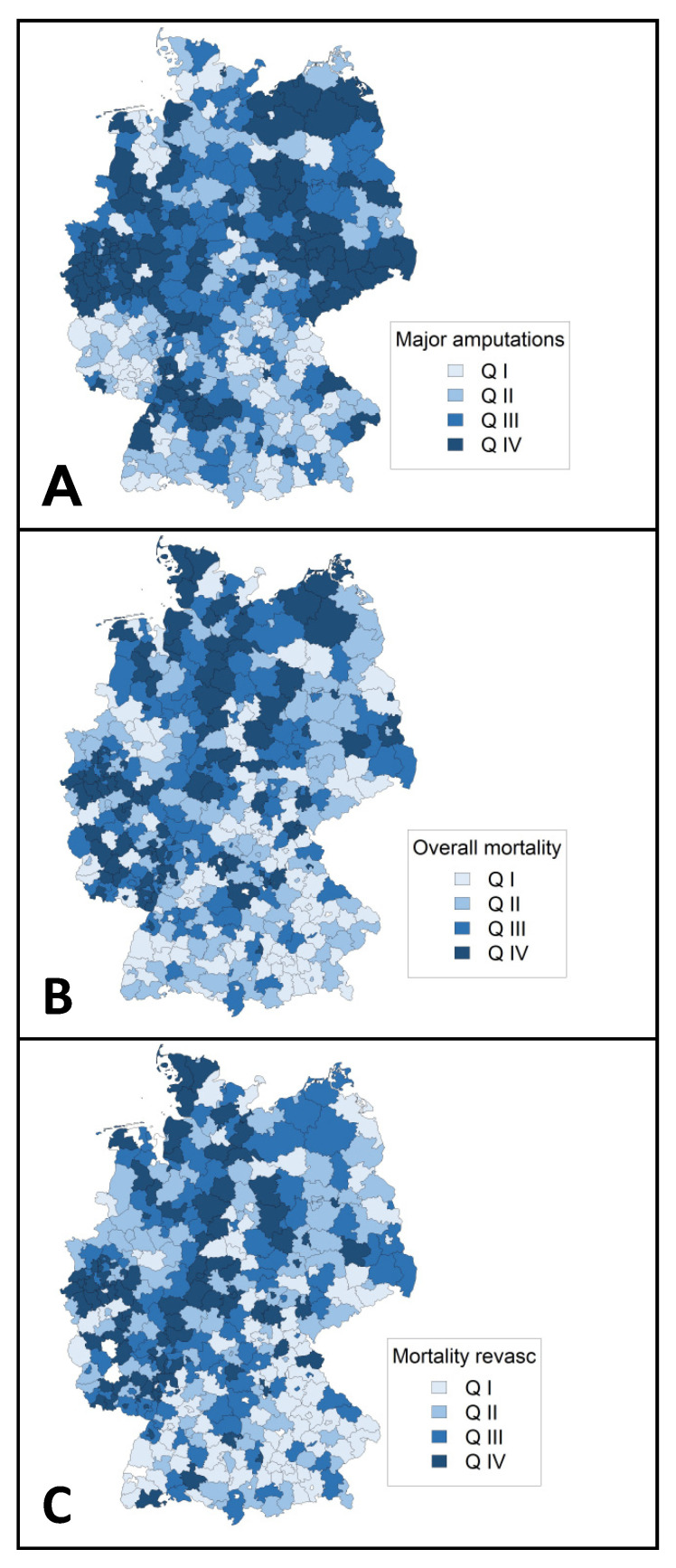
Regional distribution of outcomes (aggregated 2018) in Germany. Choropleth maps showing major amputation rate per 1000 peripheral arterial disease or arterial embolism or thrombosis cases (**A**), regional overall mortality (**B**), and regional mortality in patients who underwent revascularization (**C**). Mortalities were adjusted for age, sex, Elixhauser score, and Fontaine stage. Q indicates quartile.

**Table 1 jcm-11-02116-t001:** Characteristics of cases utilizing in-patient services for peripheral arterial disease and acute limb ischemia (2009–2018).

	Peripheral Arterial Disease or Arterial Embolism or Thrombosis		
	As MD/SD with Specific Interventional/Surgical Therapy *n* = 2,110,925	As MD without Specific Interventional/Surgical Therapy *n* = 478,586	Total *n* = 2,589,511
**Age in years** (median, Q1–Q3)	72 (63–79)	75 (66–83)	72 (63–80)
**Male sex**	1,363,772 (65)	272,242 (57)	1,636,014 (63)
**Elixhauser Score (median, Q1–Q3)**	5 (0–10)	5 (0–11)	5 (0–10)
**Comorbidities**			
Hypertension	1,509,994 (72)	326,668 (68)	1,836,662 (71)
Diabetes mellitus	652,631 (31)	155,785 (33)	808,416 (31)
Renal disease	596,071 (28)	143,048 (30)	739,119 (29)
Chronic ischemic heart disease	604,810 (29)	131,969 (28)	736,779 (29)
Chronic pulmonary disease	229,536 (11)	54,700 (11)	284,236 (11)
Aortic Aneurysm	28,479 (1.4)	8179 (1.7)	36,658 (1.4)
**Distance from residency to hospital**			
Linear distance in km (median, Q1–Q3)	9.64 (4.2–19.2)	8.81 (3.7–17.4)	9.47 (4.1–18.8)
**Treating Department**			
Vascular Surgery	655,086 (31)	114,249 (24)	769,335 (30)
General Surgery	838,201 (40)	177,838 (37)	1,016,039 (39)
Cardiology and Angiology	156,151 (7.4)	27,645 (5.8)	183,796 (7.1)
Internal medicine	299,850 (14)	93,555 (20)	393,405 (15)
Other	295,255 (14)	85,089 (18)	380,344 (15)
**Outcomes**			
Length of stay in days (median, Q1–Q3)	9 (3–18)	6 (2–12)	8 (3–17)
Annual sum of days in hospital (Q1–Q3)	6838 (4388–9540)	5690 (2767–8805)	6632 (4144–9421)
Minor amputation	302,724 (14)	-	302,724 (14)
Major amputation	146,749 (7.0)	-	146,749 (7.0)
In-hospital mortality	84,894 (4.0)	22,303 (4.7)	107,197 (4.1)

Values are given as *n* (%) and column percentages unless otherwise stated. PAD indicates peripheral arterial disease; MD, main diagnosis; SD, secondary diagnosis; *n*, number of cases; Q, quartile.

**Table 2 jcm-11-02116-t002:** Characteristics of operatively or interventionally treated cases utilizing services for lower extremity artery disease from 2009 to 2018 by main diagnosis.

Main Diagnosis	Fontaine Stage				Arterial Embolism/Thrombosis	Other
	I/IIa	IIb	III	IV		
	*n* = 38,766	*n* = 712,306	*n* = 178,608	*n* = 503,873	*n* = 141,160	*n* = 536,212
**Age in years (median, Q1–Q3)**	68 (60–75)	68 (60–75)	71 (62–79)	76 (68–83)	74 (63–83)	72 (64–79)
**Male sex**	27,019 (70)	479,908 (67)	105,804 (59)	313,052 (62)	70,840 (50)	367,149 (69)
**Elixhauser Score** **(median, Q1–Q3)**	0 (0–5)	0 (0–5)	3 (0–8)	7 (2–13)	7 (2–13)	7 (2–14)
**Comorbidities**						
Chronic ischemic heart disease	9166 (24)	178,289 (25)	51,078 (29)	151,943 (30)	32,731 (23)	181,603 (34)
Hypertension	26,589 (69)	511,423 (72)	129,352 (72)	361,501 (72)	92,389 (66)	388,740 (73)
Chronic pulmonary disease	2717 (7.0)	67,720 (9.5)	23,689 (13)	60,150 (12)	14,371 (10)	60,889 (11)
Diabetes mellitus	9820 (25)	189,046 (27)	52,480 (29)	225,326 (45)	36,813 (26)	139,146 (26)
Renal disease	5552 (14)	113,884 (16)	44,097 (25)	196,752 (39)	34,778 (25)	201,008 (38)
Aortic Aneurysm	453 (1.2)	9373 (1.3)	2806 (1.6)	5231 (1.04)	2653 (1.9)	7963 (1.5)
**Distance from residency to hospital**						
Linear distance in km (median, Q1–Q3)	11.05 (4.9–22.0)	9.88 (4.4–19.8)	9.31 (4.1–18.8)	9.38 (4.1–18.3)	9.43 (4.1–19.0)	9.59 (4.1–19.2)
**Treating Department**						
Vascular surgery	7958 (21)	223,801 (31)	59,956 (34)	167,340 (33)	49,528 (35)	146,503 (27)
General surgery	11,375 (29)	237,547 (33)	70,003 (39)	225,302 (45)	61,867 (44)	232,107 (43)
Cardiology or Angiology	6430 (17)	68,629 (9.6)	13,560 (7.6)	26,495 (5.3)	6858 (4.9)	34,179 (6.4)
General internal medicine	6074 (16)	97,597 (14)	22,383 (13)	60,523 (12)	15,936 (11)	97,337 (18)
Others	7420 (19)	97,406 (14)	21,757 (12)	64,252 (13)	19,143 (14)	85,277 (16)
**Outcomes**						
Length of stay in days (median, Q1–Q3)	2 (1–4)	3 (2–8)	8 (4–15)	14 (8–25)	9 (6–16)	15 (8–28)
Minor amputation	203 (0.52)	482 (0.07)	873 (0.49)	141,169 (28)	2379 (1.7)	157,618 (29)
Major amputation	133 (0.34)	518 (0.07)	3812 (2.1)	72,958 (15)	9363 (6.6)	59,965 (11)
In-hospital mortality	150 (0.39)	1690 (0.24)	3687 (2.1)	29,121 (5.8)	14,297 (10)	35,949 (6.7)

Values are given as *n* (%) and column percentages unless otherwise stated. *Other* comprises all cases with peripheral arterial disease or arterial embolism or thrombosis as the secondary diagnosis; *n*, number of cases; Q, quartile.

**Table 3 jcm-11-02116-t003:** Procedural characteristics of operatively treated patients utilizing services for lower extremity artery disease 2009–2018.

Main Diagnosis	Fontaine Stage				Arterial Embolism/Thrombosis	Other
	I/IIa	IIb	III	IV		
	*n* = 38,766	*n* = 712,306	*n* = 178,608	*n* = 503,873	*n* = 141,160	*n* = 536,212
**Endovascular Procedures**						
Balloon Angioplasty						
Aorta and iliac	118 (0.30)	3006 (0.42)	467 (0.26)	492 (0.10)	659 (0.47)	2617 (0.49)
Vascular Graft	196 (0.51)	2852 (0.4)	1432 (0.8)	2331 (0.46)	946 (0.67)	5925 (1.1)
Above knee	28,590 (74)	451,601 (63)	78,971 (44)	174,250 (35)	29,009 (21)	141,425 (26)
Below knee	4869 (13)	70,571 (9.9)	29,625 (17)	146,890 (29)	15,614 (11)	107,990 (20)
Stent Angioplasty						
Aorta and iliac	94 (0.24)	2092 (0.29)	339 (0.19)	364 (0.07)	485 (0.34)	1440 (0.27)
Vascular Graft	36 (0.09)	546 (0.08)	361 (0.20)	405 (0.08)	282 (0.20)	1050 (0.20)
Above knee	17,027 (44)	274,732 (39)	45,053 (25)	80,506 (16)	16,155 (11)	66,768 (13)
Below knee	838 (2.2)	10,349 (1.5)	4418 (2.5)	16,409 (3.3)	2785 (2.0)	10,342 (1.9)
**Open Surgical Procedures**						
Endarterectomy						
Aorta and iliac	1260 (3.3)	49,815 (7.0)	15,698 (8.8)	22,912 (4.6)	8107 (5.7)	23,487 (4.4)
Above knee	4010 (10)	147,251 (21)	50,291 (28)	84,075 (17)	39,324 (28)	93,037 (17)
Below knee	75 (0.19)	2284 (0.32)	4185 (2.3)	8612 (1.71)	4708 (3.3)	9114 (1.7)
Bypass (distal anastomosis)						
Iliac	33 (0.09)	1574 (0.22)	427 (0.24)	297 (0.06)	469 (0.33)	1349 (0.25)
Femoral	585 (1.5)	25,813 (3.6)	13,151 (7.4)	15,358 (3.1)	7050 (5.0)	20,686 (3.9)
Popliteal above knee	781 (2.0)	39,203 (5.5)	10,188 (5.7)	23,355 (4.6)	2180 (1.5)	13,764 (2.6)
Popliteal below knee	276 (0.71)	12,854 (1.8)	10,612 (5.9)	24,688 (4.9)	3629 (2.6)	16,758 (3.1)
Popliteal unspecified	89 (0.23)	3324 (0.47)	1354 (0.76)	2900 (0.58)	436 (0.31)	2089 (0.39)
Crural	186 (0.48)	5260 (0.74)	12,458 (7.0)	35,562 (7.1)	4658 (3.3)	22,128 (4.1)
Pedal	10 (0.03)	87 (0.01)	486 (0.3)	5289 (1.1)	310 (0.22)	3988 (0.74)
**Major Amputation**						
Above knee and knee exarticulation	101 (0.26)	403 (0.06)	3017 (1.7)	50,862 (10)	7687 (5.5)	35,539 (6.6)
Below knee	36 (0.09)	151 (0.02)	1080 (0.60)	26,063 (5.2)	2206 (1.6)	27,754 (5.2)
Unspecified		1 (0.00)	6 (0.00)	268 (0.05)	16 (0.01)	278 (0.05)
**Minor Amputation**						
Forefoot and midfoot	40 (0.10)	88 (0.01)	278 (0.16)	38,816 (7.7)	902 (0.64)	45,043 (8.4)
Toes	164 (0.42)	394 (0.06)	635 (0.36)	111,483 (22)	1580 (1.1)	121,216 (23)
Unspecified	8 (0.02)	11 (0.00)	18 (0.01)	2705 (0.54)	48 (0.03)	3679 (0.69)

Values are given as *n* (%) and column percentages unless otherwise stated. *Other* comprises all cases with peripheral arterial disease or arterial embolism or thrombosis as the secondary diagnosis; *n*, number of patients.

## Data Availability

Restrictions apply to the availability of these data. Data were obtained from the German Federal Statistical Office and are available from the authors with the permission of the German Federal Statistical Office.

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
