# Peer review of "Utilization and Regional Differences of In-Patient Services for Peripheral Arterial Disease and Acute Limb Ischemia in Germany: Secondary Analysis of Nationwide DRG Data"

_jcm, 2022, doi:10.3390/jcm11082116_

Round 1

Reviewer 1 Report

Overall:

This is a well-written and interesting report of the German healthcare system regarding lower limb obstructive disease. I see one major issue:

According to the title and main conclusions of your study, you focused on PAD patients. However, also patients with arterial embolism were included, which are not easily comparable to PAD patients: It is usually an acute ischemia with higher complications/amputation rate and also higher mortality (which corresponds with your results) and different etiology than atherosclerotic PAD. You also use the term of acute limb ischemia in the last part of the results section.

Therefore, these DRG codes should be excluded from your analysis or the focus, title and conclusion of your study should be changed to include chronic and acute occlusive diseases of the lower limbs.

Abstract

Table 1: Please consider sorting in accordance to total numbers (e.g. Hypertension first,…)

Main text

Please add the classification of acute ischemia following acute embolism according to Rutherford.

Reviewer 2 Report

Review JCM-16566543

Thank you for the opportunity to review this interesting manuscript. I think the manuscript is of relevance and interest for the reader.

I have one major comment that should be addressed from my point of view:

  • As described the treating departments are coded into vascular surgery, general surgery, cardiology/angiology and internal medicine. However, as also stated in the limitations section this is mainly a coding problem. Those coded as general surgery are originally treated by vascular units. I think this should be made clear also in the results section an in the tables, as otherwise readers who are not familiar with the German health system might think in Germany most PAD patients are treated by general surgeons.
  • Page 10 line 50: Following the comment above, I cannot fully agree with the results discussed here. Can you differentiate if the patient was treated by a vascular unit in a general surgery department (which is of course similar specified as a dedicated vascular center) or a dedicated coded separate vascular unit?

Round 2

Reviewer 1 Report

No further comments.